# The Association between Oxytocin and Lower Limb Osteoarthritis: A Prospective Cohort Study

**DOI:** 10.3390/ijms24119750

**Published:** 2023-06-05

**Authors:** Christian Hubert Roux, Anne-Sophie Rousseau, Antonio Iannelli, Nadine Gautier, Stéphanie Ferrero, Charlotte Hinault, Giulia Chinetti, Willy Ngueyon-Sime, Francis Guillemin, Ez Zoubir Amri

**Affiliations:** 1CHU, Inserm, Université Côte d’Azur, 06000 Nice, France; ferrero.s@chu-nice.fr; 2CNRS, Inserm, iBV, Université Côte d’Azur, 06000 Nice, France; anne-sophie.rousseau@univ-cotedazur.fr (A.-S.R.); nadine.gautier@univ-cotedazur.fr (N.G.); 3Centre Hospitalier Universitaire de Nice-Digestive Surgery and Liver Transplantation Unit, Archet 2 Hospital, 151 Route Saint Antoine de Ginestière, BP 3079, CEDEX 3, 06200 Nice, France; iannelli.a@chu-nice.fr; 4CHU, Inserm, C3M, Université Côte d’Azur, 06000 Nice, France; hinault.c@chu-nice.fr (C.H.); chinetti.g@chu-nice.fr (G.C.); 5CHRU de Nancy, Inserm, Université de Lorraine, CIC Clinical Epidemiology, 54500 Nancy, France; w.ngueyon-sime@chru-nancy.fr (W.N.-S.); francis.guillemin@chru-nancy.fr (F.G.)

**Keywords:** oxytocin, osteoarthritis, knee, hip, structural damages, related factors

## Abstract

Oxytocin (OT), a neuropeptide best known for its role in emotional and social behaviors, has been linked to osteoarthritis (OA). This study aimed to investigate the serum OT level in hip and/or knee OA patients and to study its association with disease progression. Patients from the KHOALA cohort with symptomatic hip and/or knee OA (Kellgren and Lawrence (KL) scores of 2 and 3) and follow-up at 5 years were included in this analysis. The primary endpoint was structural radiological progression, which was defined as an increase of at least one KL point at 5 years. Logistic regression models were used to estimate the associations between OT levels and KL progression while controlling for gender, age, BMI, diabetes and leptin levels. Data from 174 hip OA patients and 332 knee OA patients were analyzed independently. No differences in OT levels were found between the ‘progressors’ and ‘non-progressors’ groups among the hip OA patients and knee OA patients, respectively. No statistically significant associations were found between the OT levels at baseline and KL progression at 5 years, the KL score at baseline or the clinical outcomes. Higher structural damage at baseline and severe structural progression of hip and knee osteoarthritis did not appear to be associated with a low serum OT level at baseline.

## 1. Introduction

Oxytocin (OT) is a neuropeptide synthesized by the paraventricular and supraoptic nuclei of the hypothalamus and is secreted by the posterior pituitary gland into the general circulation. OT plays an essential role in reproduction. However, OT also modulates several other functions. Indeed, studies have shown that OT inhibits adipocyte formation and stimulates osteoblastogenesis [1,2] and chondrogenesis [3]. Treatment with OT prevents the degradation of the chondrocytes [3].

Osteoarthritis (OA) is the most common joint disease, affecting more than 303 million people worldwide [4]. The prevalence of osteoarthritis, calculated in individual studies, depends on the definition of osteoarthritis used, as well as the age categories, countries of origin and sex distribution of the study population [5]. With the ageing of the population in Western countries, the proportion of the population aged 45 years or older with doctor-diagnosed osteoarthritis is estimated to increase from 26.6% to 29.5% for any location, from 13.8% to 15.7% for the knee and from 5.8 to 6.9% for the hip by 2032 [6].

Osteoarthritis is a disease of the whole joint, characterized by the progressive destruction of cartilage [7]. As recommended by the EULAR, the structural changes should be assessed primarily via conventional radiography, and the use of additional, more sophisticated imaging, such as MR imaging, should be restricted to atypical cases or patients showing rapid progression of symptoms [8].

There is variability in the structural radiological progression and in the clinical progression, depending on various etiological factors. All treatments that are currently available target the symptomatic effects but are unable to delay the degenerative evolution of the disease. It remains a disease with a complex pathophysiology, and our knowledge of it is still incomplete despite the progress made in recent years. Hormonal, genetic, metabolic and biomechanical factors definitely play roles in its development [9,10,11,12,13,14].

Our team previously showed that serum OT levels were 3.4 times lower in women with hand OA compared to controls without hand OA [3]. Furthermore, in vitro, it was shown that OT could prevent the destruction of the cartilage matrix by regulating the matrix metalloproteinases [15]. To our knowledge, no other work has been conducted on the relationship between OT and OA [16]. We hypothesized that higher structural damage at baseline and severe structural progression of OA are associated with a low serum OT level. For these reasons, this longitudinal study aimed to investigate the serum OT level in hip and/or knee OA patients and to study its association with disease progression.

## 2. Results

### 2.1. Baseline Characteristics

In total, 545 participants in the KHOALA cohort were followed for 5 years, including 192 hip OA patients and 380 knee OA patients. Among them, 174 hip OA patients and 332 knee OA patients with available data regarding Kellgren and Lawrence (KL) progression were included in the present study (Figure 1).

The baseline characteristics and hormone levels of the study population are presented in Table 1. Overall, 133 hip OA patients had mild structural disease severity (69%), as was the case for 233 knee OA patients (61%).

### 2.2. Factors Associated with Serum OT Levels at Baseline

In the hip and/or knee OA patients, at baseline, no statistically significant associations were found between serum OT levels and pain (hip: *p* = 0.92; knee: *p* = 0.58) and mobility function (hip: *p* = 0.97; knee: *p* = 0.76). Only BMI was statistically associated with OT levels at baseline in the hip OA patients (linear regression adjusted for pain and mobility function, beta = 6.6, *p* = 0.009).

### 2.3. Factors Associated with the KL Score at Baseline

In the hip and/or knee OA patients, no statistically significant association was found between the KL score at baseline and each of the explored variables, including the serum OT level. Only gender was significantly associated with the KL score (OR = 0.55 {0.35–0.87]; *p* = 0.01) (Table 2).

### 2.4. Patient Characteristics according to the Kellgren and Lawrence Progression Score at 5 Years

Among the patients with hip OA, 87 patients were classified as ‘non-progressors’, and 87 patients were ‘progressors’. In the case of hip OA, we found KL 3 rates of 40.2% in progressors and 20.7% in non-progressors and KL 2 rates of 59.8% in progressors and 79.3% in non-progressors. In the case of knee OA, we found KL 3 rates of 43.7% in progressors and 38.3% in non-progressors and the KL 2 rates of 56.3% in progressors and 61.7% in non-progressors.

The serum OT levels at baseline were 3.3 [1.8–9.7] pg/mL and 2.7 [2.0–6.5] pg/mL in ‘non-progressors’ and ‘progressors’, respectively. There were no significant differences among the ‘progressors’ and ‘non-progressors’ in terms of gender (*p* = 0.15), age (*p* = 0.29), BMI (*p* = 0.13), comorbidity index (*p* = 0.24), smoking status (*p* = 0.84), Still’s disease (*p* = 0.41), and mobility function (*p* = 0.13), as well as oxytocin (*p* = 0.59), leptin (*p* = 0.08) and estradiol (*p* = 0.94) levels. The pain and adiponectin levels were significantly higher in the ‘progressors’ group than in the ‘non-progressors’ (*p* = 0.04 and 0.009, respectively) (Table 3).

Among the patients with knee OA, 206 patients were classified as ‘non-progressors’, and 126 patients were ‘progressors’. No parameter measured at baseline was statistically significantly associated with knee OA progression (Table 4).

### 2.5. Factors Associated with KL Progression

Logistic regression analyses were conducted to estimate the association between KL progression and the serum OT level (Table 5). Using multivariable models adjusted for gender, age, BMI, diabetes and estradiol and leptin levels, no significant associations were found between KL progression and the explored variables, including serum OT levels.

## 3. Discussion

Our study showed that plasma basal levels of OT in knee and hip OA patients did not differ, and the main result was that an association between OT and the radiological progression of OA could not be found.

Our understanding of the pathophysiology of osteoarthritis remains incomplete. The involvement of certain hormones remains debated. Our previous study showed that serum OT levels were significantly lower in patients with hand OA than in patients without hand OA [3]. A negative association between OT and OA exists in humans [3], suggesting a preventive role of OT in OA that is effected by increasing chondrogenesis [3]. The available data on the role of OT in OA from in vitro and animal studies also suggest the therapeutic potential of OT to prevent cartilage degradation. The oxytocin receptor (OTR) is expressed in human primary chondrocytes but is found at a lower level in the chondrocytes of OA patients [15]. OTR expression is dose-dependently reduced by TNF-α treatment, and the inhibitory effect of OT on the TNF-α-induced degradation of Col II is dependent on OTR [15]. All these elements reinforce the hypothesis of a role of oxytocin in OA. Our negative results can be explained by the complexity of the physiopathology of this disease and the multiplicity of factors involved, as well as the difference in phenotype between hand and lower limb OA.

Endogenous OT levels in OA patients may not be involved in the inter-individual variation in OA progression. However, no reference value for plasma OT exists, and it is problematic that there is very high variability in the reported OT levels between studies, even within similar populations and contexts [17,18,19]. These inconsistencies can be partly explained by different measurement methods. We used a valid and reliable method for OT dosage (e.g., an extraction procedure before assaying), as reported in [3]. Moreover, our results consider known potential confounders such as age and gender [3]. We did not find any relationship of age and gender with OA progression of the hip and/or knee, in line with the findings of a systematic review and meta-analysis evaluating prognostic factors for the radiological progression of OA [20,21].

There are many factors that can affect the death of chondrocytes, such as the release of local inflammatory factors and lipid metabolism. Considering the complex and multiple functions of OT, such as the lipolytic effect in adipose tissues, we also adjusted the OT statistical analysis based on the leptin level, BMI and diabetes. Indeed, leptin, a proinflammatory adipokine and satiety hormone secreted proportionally to adipose tissue mass, is consistently increased in obesity-induced OA [22]. In this study, no association was found between leptin and OA. This was not unexpected, as the population included in this study was not obese but overweight. Interestingly, adiponectin, an adipokine with anti-inflammatory properties acting on chondrocytes [23], was higher in the hip progressors compared to non-progressors. However, no relationship with OA progression was found. Little is known about OA pathogenesis and adiponectin, but it was later found to have an unexplained pro-inflammatory effect on rheumatoid arthritis [24].

Pain was significantly higher in the ‘progressors’ group than in the ‘non-progressors’ group among the hip OA patients alone. Independent of its potential effect on chondrocytes, OT can have a beneficial effect on OA pain, one of the clinical markers of OA progression [16]. Indeed, OT has been implicated in the modulation of somatosensory transmission, such as nociception and pain [25,26]. However, peripheral OT was not associated with pain in the OA patients at baseline. An assessment of the relationship between serum OT levels and clinical/biological OA progression, including inflammatory cytokines, is one of the most interesting points that will need to be considered in future studies. Indeed, factors associated with radiological OA progression must be distinguished from factors associated with clinical/biological OA progression, as many studies show a discrepancy between clinical and radiological features [27]. Because both symptomatic and structural variables of OA should be assessed in clinical studies, an important next step for advancing the design of trials of treatments for OA is to combine these variables into a single composite index [28].

Our study has several strengths, including a large sample size of knee and/or hip OA patients, formed of patients with predominantly mild radiological OA at baseline, as well as a long follow-up time of 5 years in a real-life setting and a robust analysis of radiographs in a unique, centralized radiology reading center. However, it has limitations regarding the oxytocin dosages and, more precisely, the timing of the dosages. These were all administered in the morning, but it is impossible to know if they were administered under the same conditions in this multicenter study and in the absence of stress. Furthermore, we cannot exclude an effect realized through OTR, such as lower OTR levels or dysfunction in downstream signaling. It is worth noting that the turnover of OT might also be considered, as it is a target of oxytocinases that are probably controlled by OA. Another limitation is that our study did not take other treatments into account, particularly opioids and especially estrogens, which could have possibly distorted the measurement of the oxytocin levels.

In this study, outcomes of hip and/or knee OA patients at baseline and structural progression at 5 years did not appear to be associated with serum OT levels. There is no evidence leading us to consider OT level as an independent biomarker of OA progression in either hip or knee OA, considering the structural variables of OA. However, further studies are needed to consider the concomitant effects of OT on multiple other tissues (subchondral bone, muscle, adipocytes, etc.) that play important roles in OA and its anti-inflammatory action and to include the symptomatic variables of OA.

## 4. Materials and Methods

### 4.1. Study Design and Data Collection

KHOALA was a French, multicenter cohort study that included patients with uni- or bilateral symptomatic hip and/or knee OA (American College of Rheumatology (ACR) criteria) and a KL score of 2 or greater. The objective, design and characteristics of the cohort have been described previously [29]. All radiographs were read centrally by two experienced readers who were blinded to the clinical conditions and questionnaire results. Single readings of radiographs were prepared after training both readers on a pilot study sample (N = 1380). Target hip and knee femoro-tibial compartments were scored using the Kellgren–Lawrence (KL) method on the basis of the degree of osteophyte formation, joint space narrowing, sclerosis, and joint deformity, distinguished according to five grades (0: no OA, 1: doubtful, 2: minimal, 3: moderate, 4: severe). Cases were defined as symptomatic OA by the physician (clinical) and OA with KL ≥ 2 on radiography for the same joint [30].

Patients with KL scores of 2 and 3 at baseline and follow-up at 5 years were included in this analysis. For the present study, data collected at baseline and at the 5-year follow-up were used.

All participants gave written informed consent to be included in the KHOALA cohort. The ethics committee CPP Est III approved the cohort study (no. 07.01.01), registered at ClinicalTrials.gov (no. NCT00481338).

The primary endpoint was structural radiological progression, which was defined as an increase of at least one KL point at 5 years.

### 4.2. Biological Analysis

At baseline, the participants underwent blood collection. The serum samples were stored at −80 °C until analysis. The serum OT levels were measured using a radioimmunoassay (RIA Phoenix Pharmaceutical kit). The solid-phase extraction of the serum samples was performed to eliminate the effects of potentially interacting molecules [31].

The serum level of estradiol, which is highly sensitive, was measured via ELISA.

The serum adiponectin and leptin levels were available in the KHOALA database.

### 4.3. Statistical Analysis

Statistical analyses were carried out using the SAS/STAT 9.04.01 software (SAS Institute Inc., Cary, NC, USA). Categorical variables are presented as numbers (percentages), while continuous variables are expressed as medians with interquartile ranges [Q1–Q3].

Comparisons between the progressor and non-progressor groups were performed using Student’s *t*-test or the Kruskal–Wallis test if heteroscedasticity was observed for continuous variables, and the chi-squared test or Fisher’s exact test was used for categorical variables.

A linear regression was used to measure dependence between the OT levels and the baseline characteristics (gender, age, occupation, BMI, smoking status, pain VAS, Groll comorbidity index, OT, leptin, adiponectin and estradiol levels, normalized pain and function). Multivariable regressions with forced adjustment for normalized pain [0,100] and normalized function [0,100] were performed. Only factors with a significant association at the 0.2 threshold in the bivariate regression were selected for the multivariable model.

Logistic regression models were constructed to estimate the association between the serum OT level and each outcome (KL progression, KL at baseline) with the odds ratio (OR) and corresponding 95% confidence interval (CI), controlling for gender, age, BMI, diabetes and estradiol and leptin levels. Only factors with an association at *p* value < 0.2 in the bivariable analysis were entered into the multivariable models. Stepwise variable selection was used, with the significance level for entry into the model at 0.2 and the significance level for staying in the model at 0.05; however, for OT, the multivariable model was used. Analyses were performed separately for hip and knee OA. A two-sided *p* value < 0.05 was considered statistically significant.

## Figures and Tables

**Figure 1 ijms-24-09750-f001:**
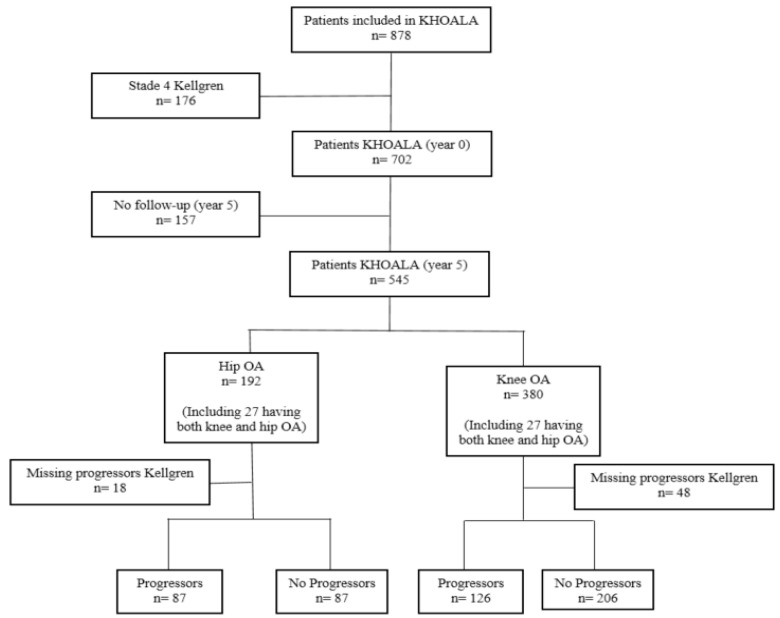
Flowchart of the study population.

**Table 1 ijms-24-09750-t001:** Patient baseline characteristics.

	Hip OA (n = 192)	Knee OA (n = 380)
Female	128 (66.7%)	267 (70.3%)
Age (years)	61.5 [56.5–68.0]	61.0 [55.0–69.0]
BMI (kg/m^2^)	26.3 [24.3–29.4]	28.8 [25.7–32.7]
Groll comorbidity index	2.0 [1.0–3.0]	2.0 [1.0–3.0]
Smokers	30 (15.6%)	53 (14.1%)
Still’s disease	6 (3.1%)	14 (3.7%)
Normalized Pain VAS (0–100)	30.0 [20.0–45.0]	30.0 [15.0–40.0]
Normalized function (0–100)	31.3 [14.8–49.8]	28.7 [13.1–45.9]
Kellgren and Lawrence score		
2	133 (69.3%)	233 (61.3%)
3	59 (30.7%)	147 (38.7%)
Oxytocin (pg/mL)	3.0 [1.8–7.2]	3.1 [1.6–6.7]
Adiponectin (μg/mL)	5.6 [3.9–8.0]	5.8 [4.0–8.2]
Leptin (ng/mL)	12.8 [6.4–27.8]	22.0 [9.2–44.6]
Estradiol (pg/mL)	11.5 [11.0–21.6]	12.4 [11.0–21.8]

Data are presented as the median [Q1; Q3] or number (%). BMI: Body Mass Index; OA: osteoarthritis; VAS: Visual Analogical Scale.

**Table 2 ijms-24-09750-t002:** Factors associated with the KL score at baseline in hip and/or knee OA patients, derived from bivariable regression.

	Hip OA			Knee OA		
	OR	95% CI	*p* Value	OR	95% CI	*p* Value
Gender: female	0.96	0.50–1.84	0.91	0.55	0.35–0.87	0.01
Age at baseline	1.14	0.78–1.67	0.49	1.07	0.84–1.35	0.60
BMI (kg/m^2^)	0.73	0.35–1.52	0.40	1.29	0.88–1.87	0.19
Smokers	1.63	0.73–3.65	0.24	1.39	0.77–2.49	0.27
Normalized pain VAS	1.03	0.90–1.17	0.68	1.07	0.98–1.16	0.16
Groll comorbidity index	0.93	0.75–1.15	0.50	1.07	0.93–1.23	0.35
Oxytocin (pg/mL)			0.23			0.31
1st decile	1			1		
2nd decile	0.10	0.01–0.94		0.95	0.32–2.79	
3rd decile	0.48	0.10–2.30		1.10	0.38–3.20	
4th decile	1.17	0.27–5.05		0.95	0.32–2.79	
5th decile	1.00	0.22–4.47		0.69	0.23–2.11	
6th decile	0.48	0.10–2.30		1.66	0.58–4.75	
7th decile	0.89	0.20–3.90		1.45	0.51–4.16	
8th decile	0.89	0.20–3.90		1.66	0.58–4.75	
9th decile	0.33	0.06–1.73		1.66	0.58–4.75	
>9th decile	0.36	0.07–1.91		2.85	0.99–8.21	
Adiponectin (μg/mL)			0.33			0.25
1st decile	1			1		
2nd decile	2.92	0.62–13.7		0.95	0.38–2.36	
3rd decile	4.50	0.97–20.8		0.69	0.27–1.73	
4th decile	2.31	0.48–11.1		1.31	0.53–3.24	
5th decile	1.79	0.36–8.90		0.90	0.36–2.24	
6th decile	2.92	0.62–13.8		0.42	0.16–1.11	
7th decile	0.59	0.09–4.01		0.69	0.27–1.73	
8th decile	3.33	0.72–15.4		0.86	0.34–2.13	
9th decile	2.50	0.51–12.1		0.61	0.24–1.55	
>9th decile	2.31	0.48–11.1		0.38	0.14–1.02	
Leptin (ng/mL)			0.66			0.17
1st decile	1			1		
2nd decile	2.04	0.48–8.71		2.31	0.91–5.91	
3rd decile	1.25	0.28–5.65		1.69	0.66–4.32	
4th decile	2.55	0.61–10.7		0.56	0.20–1.57	
5th decile	0.93	0.20–4.47		1.42	0.55–3.68	
6th decile	3.15	0.75–13.2		1.69	0.66–4.32	
7th decile	1.62	0.37–7.05		1.22	0.47–3.15	
8th decile	2.04	0.48–8.71		0.85	0.32–2.27	
9th decile	1.25	0.28–5.65		1.36	0.53–3.50	
>9th decile	0.93	0.20–4.47		1.77	0.69–4.55	
Estradiol > 11 pg/mL	0.89	0.45–1.75	0.73	1.24	0.79–1.95	0.35

BMI: Body Mass Index; CI: confidence intervals; OA: osteoarthritis; OR: odds ratio; VAS: Visual Analogical Scale.

**Table 3 ijms-24-09750-t003:** Hip OA patient characteristics according to the Kellgren and Lawrence progression score at 5 years.

	Hip OA (n = 174)		
	Non-Progressors (n = 87)	Progressors (n = 87)	* *p* Value
Female	52 (59.8%)	61 (70.1%)	0.15
Age at baseline	61.0 [55.0–68.0]	62.0 [58.0–68.0]	0.29
BMI (kg/m^2^)	25.4 [23.9–29.0]	27.4 [24.8–29.4]	0.13
Groll comorbidity index	2.0 [1.0–3.0]	1.0 [1.0–3.0]	0.24
Smokers	15 (17.2%)	14 (16.1%)	0.84
Still’s disease	2 (2.3%)	4 (4.6%)	0.41
Normalized pain VAS (0–100)	25.0 [15.0–40.0]	30.0 [20.0–50.0]	**0.04**
Normalized function (0–100)	26.2 [13.1–47.5]	33.6 [21.3–49.8]	0.13
Oxytocin (pg/mL)	3.3 [1.8–9.7]	2.7 [2.0–6.5]	0.59
Adiponectin (μg/mL)	4.9 [3.4–6.9]	6.3 [4.2–8.6]	**0.009**
Leptin (ng/mL)	9.9 [5.9–22.9]	16.7 [6.4–32.2]	0.08
Estradiol (pg/mL)	12.2 [11.0–22.8]	11.0 [11.0–20.7]	0.94

Data are presented as the median [Q1; Q3] or number (%). * *p* value vs. non-progressors. BMI: Body Mass Index; OA: osteoarthritis; VAS: Visual Analogical Scale.

**Table 4 ijms-24-09750-t004:** Knee OA patient characteristics according to the Kellgren and Lawrence progression score at 5 years.

	Knee OA (n = 332)		
	Non-Progressors (n = 206)	Progressors (n = 126)	* *p* Value
Female	141 (68.4%)	85 (67.5%)	0.85
Age at baseline	60.0 [54.0–67.0]	61.0 [56.0–68.0]	0.13
BMI (kg/m^2^)	28.0 [25.4–31.6]	29.4 [26.1–34.0]	0.06
Groll comorbidity index	2.0 [1.0–3.0]	2.0 [1.0–3.0]	0.72
Smokers	25 (12.3%)	22 (17.6%)	0.18
Still’s disease	5 (2.4%)	8 (6.3%)	0.07
Normalized pain VAS (0–100)	25.0 [15.0–40.0]	30.0 [15.0–45.0]	0.28
Normalized function (0–100)	27.9 [11.5–45.3]	29.5 [13.1–44.3]	0.77
Oxytocin (pg/mL)	3.1 [1.6–7.0]	3.0 [1.4–6.0]	0.54
Adiponectin (μg/mL)	6.1 [3.9–8.0]	5.8 [4.1–9.5]	0.42
Leptin (ng/mL)	20.1 [9.6–37.7]	21.2 [8.0–49.8]	0.66
Estradiol (pg/mL)	12.1 [11.0–21.4]	14.5 [11.0–23.1]	0.28

Data are presented as the median [Q1; Q3] or number (%). * *p* value vs. non-progressors. BMI: Body Mass Index; OA: Osteoarthritis; VAS: Visual Analogical Scale.

**Table 5 ijms-24-09750-t005:** Factors associated with the KL progression score at 5 years in hip and knee OA patients, derived from bivariable regression.

	Hip OA			Knee OA		
	OR	95% CI	*p* Value	OR	95% CI	*p* Value
Gender: female	1.58	0.84–2.96	0.15	0.96	0.60–1.54	0.85
Age at baseline	1.22	0.84–1.76	0.29	1.22	0.94–1.59	0.13
BMI (kg/m^2^)	1.76	0.85–3.64	0.16	1.46	0.98–2.17	0.06
Smokers	0.92	0.42–2.04	0.84	1.53	0.82–2.85	0.18
Normalized pain VAS	1.15	1.01–1.30	0.03	1.15	1.04–1.27	0.01
Groll comorbidity index	0.89	0.72–1.09	0.24	1.03	0.88–1.20	0.71
Oxytocin (pg/mL)			0.24			0.66
1st decile	1			1		
2nd decile	0.40	0.07–2.18		0.59	0.20–1.73	
3rd decile	1.80	0.35–9.40		0.38	0.13–1.15	
4th decile	2.00	0.39–10.3		0.47	0.16–1.40	
5th decile	0.86	0.16–4.47		0.87	0.30–2.47	
6th decile	1.00	0.20–5.07		0.53	0.18–1.53	
7th decile	1.33	0.26–6.81		0.69	0.24–2.00	
8th decile	0.50	0.10–2.58		0.50	0.17–1.49	
9th decile	0.36	0.07–1.97		0.33	0.11–1.03	
>9th decile	1.60	0.30–8.49		0.69	0.24–2.00	
Adiponectin (μg/mL)			0.40			0.07
1st decile	1			1		
2nd decile	1.09	0.27–4.41		1.00	0.36–2.81	
3rd decile	2.50	0.65–9.65		1.15	0.41–3.18	
4th decile	1.09	0.27–4.41		0.82	0.28–2.38	
5th decile	1.78	0.45–6.97		2.15	0.80–5.80	
6th decile	2.50	0.65–9.65		1.08	0.40–2.93	
7th decile	3.43	0.89–13.3		0.66	0.23–1.89	
8th decile	2.50	0.65–9.65		0.58	0.19–1.78	
9th decile	3.00	0.72–12.5		2.73	1.00–7.41	
>9th decile	4.00	0.94–17.1		1.34	0.49–3.63	
Leptin (ng/mL)			0.20			0.35
1st decile	1			1		
2nd decile	1.25	0.34–4.64		0.63	0.24–1.62	
3rd decile	0.52	0.13–2.02		0.44	0.16–1.19	
4th decile	0.51	0.12–2.12		0.49	0.18–1.29	
5th decile	1.27	0.33–4.87		0.77	0.29–2.03	
6th decile	1.88	0.47–7.53		0.31	0.11–0.88	
7th decile	0.61	0.16–2.43		0.53	0.19–1.45	
8th decile	2.06	0.52–8.18		0.81	0.30–2.16	
9th decile	1.77	0.46–6.78		1.06	0.41–2.72	
>9th decile	2.47	0.60–10.3		0.71	0.26–1.89	
Estradiol > 11 pg/mL	0.92	0.48–1.77	0.81	1.39	0.86–2.24	0.18

BMI: Body Mass Index; CI: confidence intervals; OA: osteoarthritis; OR: odds ratio; VAS: Visual Analogical Scale.

## Data Availability

Data are located in the Clinical Investigation center (Nancy, France).

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
