# Peer review of "The Association between Oxytocin and Lower Limb Osteoarthritis: A Prospective Cohort Study"

_ijms, 2023, doi:10.3390/ijms24119750_

Round 1
Reviewer 1 Report
General comments
The article The association between oxytocin and lower limbosteoarthritis:3 a prospective cohort study is well written and well organized. The purpose of this study was to examine the serum OT level in patients with hip and/or knee OA and to investigate its relationship to disease progression. The methodology of the adopted method was described in a concise and clear manner. The authors present a research paper, and the literature list contains only 28 items. In my opinion, this is far too few for this type of article. Conclusions are presented in a concise manner. In my opinion, it seems necessary to make minor corrections to allow a more thorough understanding of the topic. The following are my comments.
What was the number of KL 2 vs 3 in groups of progressors in non-progressors ?
Who evaluated the x-ray and how ?
Minor comments:
Introduction
The introduction is quite short and contains only 11 items of literature. I ask that the authors expand this chapter to highlight the research problem and its significance more.
The incidence of osteoarthritis is influenced by many factors, such as work, sports participation, musculoskeletal injuries, obesity and gender. Information about this, along with the necessary literature, should be added to the first paragraph of the introduction. Authors may find some useful information in the works: DOI 10.1016/S0140-6736(19)30417-9; DOI10.3390/app11041552; https://doi.org/10.1136/annrheumdis-2013-204763; DOI 10.3390/app10238312; https://doi.org/10.4081/or.2014.5188;
The introduction should be expanded with brief information on typical diagnostic methods (CT, X-ray, ultrasound) including physical examination, as well as alternative methods such as vibroarthrography. Authors may find some useful information in the works: https://doi.org/10.1016/j.cpet.2018.08.004; https://doi.org/10.1111/j.1617-0830.2006.00063.x; DOI 10.3390/app9194102; https://doi.org/10.1016/j.berh.2016.09.007; doi: 10.35784/acs-2022-14;
Results and Discussion
The results have been described in an improved way, allowing easy interpretation. The chapters do not need improvement.
After making the appropriate additions, the article may be accepted for publication.
The article is written correctly, the language style is good.
Author Response
Dear Sir, thank you very much for your comments. As you asked, the introduction has been expanded. So we hope that this chapter highlight the research problem and its significance as you asked. Data have been added on prevalence diagnostic method. References in link with this add have been implemented. Modifications have been done page 1, line 38-48
Who evaluated the X-ray and how ?
Thank you very much for this important question, modifications have need done page 11 line 22-230
Information on X ray evaluation have been added page 11. All radiographs reading has been centralized. All subjects underwent radiography to obtain weight-bearing anteroposterior (AP), posteroanterior semiflexed and axial/sky views of the knee and AP pelvis and oblique (Lequesne) views of the hip. Radiography was not performed in those who had one of correct quality in the past 12 months. All radiographs were read centrally by two readers who were blinded to clinical condition and questionnaire results. Single reading of radiographs was prepared by training both readers on a pilot study sample (N = 1380). Target hip and knee femoro-tibial compartments were scored by the Kellgren–Lawrence (K–L) method on the basis of the degree of osteophyte formation, joint space narrowing, sclerosis, and joint deformity distinguished in five grades (0: no OA, 1: doubtful, 2: minimal, 3: moderate, 4: severe). Cases were defined as symptomatic OA according to the physician (clinical) and OA with K–L ≥ 2 on radiography for the same joint.
What was the number of KL2 vs 3 in groups of progression in non-progressors ?
Table 6 describes the number of Kellgren and Lawrnce score at baseline in group and for OA joint:
|
Table 6: Number of KL in group of KL progression score at 5-years |
||||||||
|
|
Hip OA |
Knee OA |
||||||
|
(n= 174) |
(n= 332) |
|||||||
|
Non-progressors |
Progressors |
Non-progressors |
Progressors |
|||||
|
(n=87) |
(n=87) |
(n=206) |
(n=126) |
|||||
|
Kellgren and Lawrence score at baseline |
|
|
|
|
|
|
|
|
|
Stade 2 |
69 |
(79.3%) |
52 |
(59.8%) |
127 |
(61.7%) |
71 |
(56.3%) |
|
Stade 3 |
18 |
(20.7%) |
35 |
(40.2%) |
79 |
(38.3%) |
55 |
(43.7%) |
|
Data are presented as number and percentage |
||||||||
In Hip OA, we have in KL 3 rate of 40.2% in progressors and 20.7% in non-progressors, and in KL 2 rate of 59.8% in progressors and 79.3% in non-progressors.
Elsewhere, in Knee OA, we have in KL 3 rate of 43.7% in progressors and 38.3% in non-progressors, and the KL 2 rate of 56.3% in progressors and 61.7% in non-progressors.
Modifications have been done page 6, line 101-104

Reviewer 2 Report
The major criticism relates to the description of the KHOALA cohorts. For example the reader is not aware of concomitant medication such as opioids and E2 therapy for menopause, potential influence of diurnal variation on OT levels, and other subgroup analysis of various phenotypes since only group data is presented. Conclusions would be more assured with sensitivity analysis represented with forrest plot.
Aside from a few grammatical errors the quality of English is fine and easy to read.
Author Response
Thank you for your very interesting and accurate comment. Indeed, different parameters can influence oxytocin levels. Some of these factors have been taken into account, but opioid and estrogen intake have not been taken into account. This is a limitation of the study and we have added it as a limitation in the discussion. The khoala cohort has been well characterised [Guillemin F, Rat A-C, Roux CH, Fautrel B, Mazieres B, Chevalier X, et al. The KHOALA cohort of knee and hip osteoarthritis in France. Joint Bone Spine 2012;79:597–603. https://doi.org/10.1016/j.jbspin.2012.03.011.] but these data are not available.
Modification has been done page 11, line 206-208
The idea of a subgroup analysis taking into account the different phenotypes is indeed excellent. However, it is difficult to envisage a subgroup analysis, because in this type of analysis we necessarily lose power. And since our results show no association, it is more than certain that with less power we would find no association. It would be an excellent idea if we had the possibility to work on data from other cohorts and in particular by combining data from several cohorts.
